# How Does the Provision of Shade during Grazing Affect Heat Stress Experienced by Dairy Cows in Sweden?

**DOI:** 10.3390/ani13243823

**Published:** 2023-12-11

**Authors:** Per Peetz Nielsen, Ewa Wredle

**Affiliations:** 1Department of Agriculture and Food, Division of Bioeconomy and Health, RISE Research Institute of Sweden, RISE Ideon, 223 70 Lund, Sweden; 2Department on Animal Nutrition and Management, Swedish University of Agricultural Sciences, 750 07 Uppsala, Sweden; ewa.wredle@slu.se

**Keywords:** heat stress, dairy cattle, access to shade, milk production, Scandinavian conditions

## Abstract

**Simple Summary:**

Heat stress in dairy cattle is well-studied under warm climates but less is known in a temperate climate such as in Sweden. This study explores the impact of heat stress on high-yielding dairy cows during the summer, with a particular focus on the climate in Scandinavia. When cows are in a state of negative energy balance during early lactation, they become more susceptible to the adverse effects of heat stress, and this can affect their milk production. The temperature humidity index (THI) is used to assess the degree of heat stress, with a THI of 72 considered the minimum borderline for when heat stress becomes a problem. As THI increases, daily milk yield decreases, showing a clear link between heat stress and reduced milk production, and providing shade proved in this study to be effective in countering the adverse effects of heat stress, especially for cows in early lactation. Those with shade access experienced smaller declines in milk production compared to those without it. In summary, this research underscored the importance of shade provision especially for early-lactation dairy cows during the summer to maintain high milk production and mitigate the adverse effects of heat stress.

**Abstract:**

Heat stress in dairy cows can cause an increase in body temperature and respiration rate, and a decreased feed intake leading to reduced production. Dairy cows are better at handling heat when they have access to shade. Therefore, this study aimed to determine the effects of providing shade to high-yielding dairy cattle during the summer in the Swedish climate. Twenty high-yielding Swedish Red dairy cows, held on pasture, were divided into two groups, one with access to shade (S) and one without (NS). Milk production was recorded daily and shade temperature and relative humidity were recorded at 10 min intervals at pasture. A major effect of heat stress was found in cows in early lactation in the NS group. In this group, a high mean temperature two days before and a high THI two days before affected the milk production negatively (*p* < 0.001), which was the same for the maximum temperature and maximum THI measured on the same day (*p* < 0.001). Increases in the mean temperature and THI two days before also affected milk production negatively (*p* < 0.05) for cows in early lactation in the S group, though to a lesser extent. This study suggests that dairy cows in early lactation benefit from access to shade during summer.

## 1. Introduction

High-yielding cows are in a negative energy balance until around six to nine weeks of lactation [1], and in this more vulnerable part of cows′ lactation period, increased effects of heat stress are reported [2,3,4]. Heat stress occurs when a combination of conditions in the environment makes the effective temperature higher than the thermo-neutral zone of the cow, affecting its behaviour, physiology, and milk production. For high-producing dairy cows, reported thermo-neutral zones are between −5 and 25 °C [5]. Cows have been selectively bred for increased milk yield, resulting in doubled milk production over the last 50 years, and as a consequence they possess a much higher metabolism [6]. Therefore, it is expected that the upper border of the thermo-neutral zone might even be lower these days. Dairy cows exposed to heat stress experience a decrease in milk production, as well as an increase in body temperature and respiration rate [7,8,9].

The temperature–humidity index (THI) is the most common guideline used for measuring the degree of heat stress dairy cows are exposed to [10]. A THI of 72 is considered the borderline for heat stress for high-producing dairy cows [7,10], and daily milk yield decreases at about 0.2 kg per unit increase in THI [11]. High-yielding cows are more sensitive to thermal stress caused by warm weather than lower-yielding cows [3]. Furthermore, the stage of lactation affects the response to heat stress and some studies have shown that mid-lactation cows seem to be more sensitive to warm and humid climates compared to early- or late-lactation cows with a higher decrease in milk yield as a response to warm weather [12,13].

Dairy cows on pasture are more capable of coping with heat if they have access to shade [9,14,15], have been shown to prefer shade over sprinklers [16], and can even distinguish between different levels of shade’s ability to reduce solar radiation [9]. Positive relationships have been found between solar radiation levels [9,17], air temperature [9,18], and shade use by dairy cows, with a peak in shade use around the mid-afternoon [7]. However, just because there is access to shade does not mean that cows use it. Tucker et al. [17] reported that cows with free access to shade used it for 3.3 h out of 15.5 h of daylight when shade blocked 99% of solar radiation, and Kendall et al. [7] showed that cows with shade access produced 0.5 kg more milk than cows without it, and though most studies reported no changes in milk composition, they found higher lactose concentrations and lower fat and protein concentrations in the milk of cows that had access to shade and cooling compared to cows without this access [19]. In a previous study by Nielsen and Wredle [20], it was observed that cows during warm days and with the possibility to choose to stay indoors during the warmest part of the day opted to go to the pasture during the night.

Cows’ respiration rate has been shown to increase with an increasing environmental temperature, as panting is one of the primary cooling mechanisms in cows [21]. Giving cows access to shade, therefore, lowers the respiration rate, and combining access to shade with cooling with sprinklers, the respiration rate has been shown to be reduced by 67% compared to cows not in the shade [22]. Furthermore, lower mean body temperature [22] or the lowest body temperature [17] was found in cows with access to shade compared to cows without it, as shade protects the cows from direct solar radiation.

Dairy cows have a limited possibility to change their circadian rhythm due to a high metabolism for milk production, and the total time spent lying or grazing [7,17], standing without grazing over 24 h [7], or the general time budget [18] are not influenced by shade provision. Though the total time budget does not seem to be influenced by providing shade, a shift in grazing behaviour patterns within the day has been found. Between 12.00 and 14.00 cows with access to shade showed decreased grazing behaviour compared to cows without access to shade (19 vs. 44%), resulting from a peak in shade use at that time [7]. By contrast, the grazing behaviour of a group with access to shade was higher between 01:00 and 02:00, when the cows compensated for the lower level of grazing behaviour during the warm part of the day [7]. The main behaviour performed in the shade was reported to be standing, accounting for 2.9 out of 3.3 h of total shade use per 15.5 h of daylight [17], probably due to a conflict between the size of the shaded area and the number of cows trying to fit in the shade.

Swedish law requires farmers to give their female cattle that are older than six months access to pasture for two to four months during the summer, depending on the region, and that cattle that are kept outside during the winter in Sweden shall have access to a shelter, providing protection against the environment [23]. However, no legislation exists on shelter provision during the summer. In general, cows can cope better with cold than heat [24], and with possible warmer summers due to climate change, it could be important that cows also have access to shade during the summer months when out on pasture, even though studies have shown that cows prefer to be indoors during the summer period even in a Nordic county such as Sweden [20,25,26].

Therefore, this study aimed to examine the effects on milk yield of providing shade to lactating dairy cattle during the summer in the Swedish climate.

## 2. Materials and Methods

### 2.1. Animals

This study was conducted at Kungsängen Research Centre in Uppsala of the Swedish University of Agricultural Sciences. Twenty high-yielding Swedish Red dairy cows were held on pasture from 1 June until 27 August 2010. These months were chosen because they represent the normal grazing period in a large part of Sweden. Water from a large water trough and access to pasture were available ad libitum. Kentucky bluegrass (*Poa pratensis*) and meadow fescue (*Festuca pratensis*) were the main grass species grown on the pasture. Twice a day at around 7 a.m. and 4 p.m., the cows were fetched from the paddocks and herded to the barn. The cows were milked in a stanchion barn (DeLaval International, Stockholm, Sweden) while being fed concentrates and roughage according to their milk yield [27]. Throughout the day and night, the cows were always outside on their designated pasture, with or without access to shade as per the experimental design. Each paddock was divided into four smaller paddocks of equal size for rotational grazing. The cows were regularly moved between these smaller paddocks to maintain a consistent supply of feed.

The cows were randomly assigned into two groups of 10 cows each. One group had free access to shade (S), provided by a sixteen-sided tent with a shade cloth creating an area of 78 m² of shade and blocking 100% of solar radiation located in the paddock. A large wooden pole supported the tent in the centre while smaller wooden poles were placed around the perimeter. The tent was positioned in the centre of the paddock to provide shade to all four sections. Electrical fencing was in place on the sides of the tent that faced the smaller paddocks, preventing the cows from accessing those areas. The other group had no access to shade (NS) in their paddock. At the start of this study, the cows in the S and NS groups were on average 119.1 ± 92.7 and 142.5 ± 94.4 days in milk (DIM), with lactation numbers 1 to 5 (1 = 5, 2 = 3, 3 = 8, 4 = 3, and 5 = 3), and had an average body weight of 627.7 ± 39.7 and 614.6 ± 56.9 kg, respectively.

The lactation stage was dichotomously modelled, where 1 = DIM ≤ 49 and 2 = DIM > 49. The total number of milk yield observations for different lactation stages in the treatment groups is presented in Table 1.

### 2.2. Measurements

Milk yield was recorded for every milking and data were stored in a central database at the research farm. To measure ambient shade temperature and relative humidity, a portable weather station (HOBO U12 Temp/RH/2 External Data Logger, U12-013, Onset Computer Corporation, Bourne, MA, USA), was used. Temperature and relative humidity were recorded at 10 min intervals throughout the whole experiment. Data from the temperature and humidity logger were manually downloaded once a week throughout the experiment and stored on a local server. Based on these data, THI was calculated with 10 min intervals as follows:THI = (1.8 × T + 32) − ((0.55 − 0.0055 × RH) ∗ (1.8 × T − 26))
where T = air temperature in shade (°C) and RH = relative humidity [17]. Mean and maximum THI were calculated per calendar day.

### 2.3. Statistical Analysis

For the statistical analysis of milk yield data, a restricted maximum likelihood (REML) model in SAS 9.1 [28] was used, using the following model:Y_ijkl_ = µ + α_i_ + β_j_ + γ_k_ + AB_ik_ + BC_kl_ + ABC_ikl_ + e_ijkl_i = 1, 2, j = 1, 2, 3, 4, 5
where µ is the overall mean effect, α the treatment group effect, β the effect of the lactation number, γ the effect of THI or temperature (°C), AB is the interaction between the treatment group and THI or temperature, BC the interaction between the lactation stage and THI or temperature (°C), ABC is the three-way interaction between the treatment group, lactation stage, and THI or temperature, and e is the random measurement error. Values of THI and temperature two days before were also used in the model. TreatmentGroup * LactationNr * Cow was used as the random statement with an autoregressive correlation structure, and the experimental unit was the individual dairy cow. Since the lactation stage of a dairy cow can change during the study, it was not included as a separate factor in the model. A *p*-value of <0.05 was considered significant. The effects of temperature and THI on milk yield in the different treatment groups in different lactation stages and on different lactation stages in general were investigated. Since all the effects of the environmental factors were covered by these three- and two-way interactions, no overall effects were looked for.

## 3. Results

The effects on milk yield of the two- and three-way interactions between groups, lactation stages, and environmental conditions are presented below. No overall statistical differences between the treatment groups were found and are therefore not presented further.

### 3.1. Environmental Conditions

Daily min, mean, and max temperatures as well as the min, mean, and max THI for the whole experiment are shown in Figure 1 and Figure 2, respectively.

Mean daily temperature values for the three experimental months are displayed in Table 2. July was the warmest month with an average mean temperature of 20.9 ± 2.9 °C over 24 h and 24.0 ± 4.0 °C during daytime, from 08:00 until 20:00. Since a temperature of 25 °C is considered to be an upper border of the thermos-neutral zone, the number of days with a mean and maximum daytime temperature of more than 25 °C was calculated, showing the most warm days in July (14 and 22, respectively).

The highest values for THI were in July, with 66.9 ± 3.8 on average for 24 h periods and 70.7 ± 4.8 on average during daytime (Table 3). Since a THI of 72 is considered to be the borderline for heat stress, the number of days with a mean and maximum THI of more than 72 was calculated too, showing the most days with high THI in July (15 and 22, respectively).

### 3.2. Milk Yield

Throughout the study, the cows in the S and NS groups produced on average 31.7 ± 8.3 and 32.4 ± 9.2 kg of milk, respectively (Table 4). Since the cows were supposed to be changed during the experiment and no cows calved, cows in lactation stage 1 are only present in the first month of the experiment, June.

### 3.3. Interactions between Groups, Lactation Stages, and Environmental Conditions

Cows in the NS group in early lactation (stage 1) were more affected by increased temperatures and THI (Table 5). In this group, a significant decrease (*p* < 0.001) in milk yield was found for the mean temperature two days before (0.36 ± 0.09 kg per increase of 1 °C) and the mean THI two days before (0.14 ± 0.04 kg per increase of 1 °C), and for the maximum temperature and THI on the day itself (0.22 ± 0.05 kg per increase of 1 °C). Increases in the mean temperature and THI two days before also caused significant decreases (*p* < 0.05) for cows in lactation stage 1 in the S group, though it was smaller (0.15 ± 0.08 and 0.08 ± 0.04 kg per increase of one unit, respectively). Cows in lactation stage 2 were less affected by environmental conditions (Table 5).

### 3.4. Interactions between Lactation Stages and Environmental Conditions

When investigating the two lactation stages and not considering treatment groups, increased temperatures and THI affected cows in early lactation with a higher decrease in milk yield than cows in lactation stage 2 (Table 6).

## 4. Discussion

Increased temperatures and increased temperature–humidity index (THI) were associated with decreases in daily milk production. This was found both when temperatures were expressed as mean values for a 24 h period and as maximum values on a day, though stronger effects were associated with mean values than with maximum values. This could be explained by the fact that on days with a high mean temperature or high mean THI, these recorded values were also high during the night, which did not enable the cows to cool down during the night. High maximum ambient temperature values can be caused by a peak during the hot part of the day, but when the following night is cooler, the cow′s body temperature will decrease during the night, cooling down the cow.

The strongest effects of increasing temperature and THI on milk production were found in cows in early lactation. Increased feed intake and the elevated metabolism in order to produce enough milk can cause a rise in body temperature [2], which makes it more likely for these cows to decrease their feed intake, and thereby milk production, on hot days to prevent their body temperature from increasing too much. Increased incidences of diseases, such as clinical mastitis [29], can be another indication of increased sensitivity to heat stress. All of this combined is in line with the earlier findings that cows in early lactation suffer more from heat stress [2] and the outcomes of this study are bringing more data to support the effect that heat stress has on dairy cows, especially during the early lactation period.

The results from this study show that access to shade could decrease the potential negative effect on milk production even though the THI in Sweden rarely reaches a level of concern. Cows in early lactation with access to shade showed smaller decreases in milk production than those without access to shade, indicating that providing shade to dairy cattle can effectively decrease the severity of heat stress from a milk production perspective. Cows in later lactation in the NS group also showed decreased milk production in several situations, though these cows in the S group only showed decreased milk production for increased THI two days before. This decreasing effect of shade on lowered milk production due to heat stress is in line with prior research [7,30], concluding that cows with access to shade produced more milk than cows without it. However, these studies were performed under a warmer climate, in New Zealand, and showed a higher advantage in relation to milk production when providing cows with shade. When primiparous cows were feed-deprived, the real dip in milk production occurred some days after feed deprivation started [31]. Therefore, it is expected that a potential decrease in feed intake due to heat stress will also be directly observed in milk production two days later.

In the present study, a limited amount of data was available for cows in early lactation due to the experimental setup of not changing cows during the experiment. Therefore, it can be recommended in the future to use more cows in early lactation, since the biggest effects of increasing temperature and humidity were found in this group. In this study, July was the warmest month with the highest THI, but also a month with a low number of experiment days with cows early in lactation, and it can therefore be expected that the effects on high temperature found in this study, with high possibility, can be replicated in future studies examining more dairy cows early in lactation during warm periods. Additionally, measuring the time spent in the shade and physiological heat stress parameters such as body temperature [7,9] and respiration rate [8] can provide valuable additional information regarding the severity of heat stress, and these parameters should be considered to be included in future studies.

## 5. Conclusions

From this study, it can be concluded that early-lactation cows benefit from having access to shade during the summer months, even though the temperature rarely exceeds the temperature comfort zone of dairy cattle under Swedish weather conditions, as providing shade resulted in a significantly smaller decrease in milk production compared to cows without access to shade. By comparing the mean shade temperature two days before, it was found that cows with access to shade had a lower decrease in milk production compared to cows without access to shade. This indicates that providing shade to early-lactation cows on warm days can improve milk production even under Swedish weather conditions. Furthermore, even though the Swedish Animal Protection Act’s goal is to improve animal welfare by mandatory access to pasture or other outdoor areas during the summer months, a suggestion might be to only allow dairy cows to be outside at night during warm periods.

## Figures and Tables

**Figure 1 animals-13-03823-f001:**
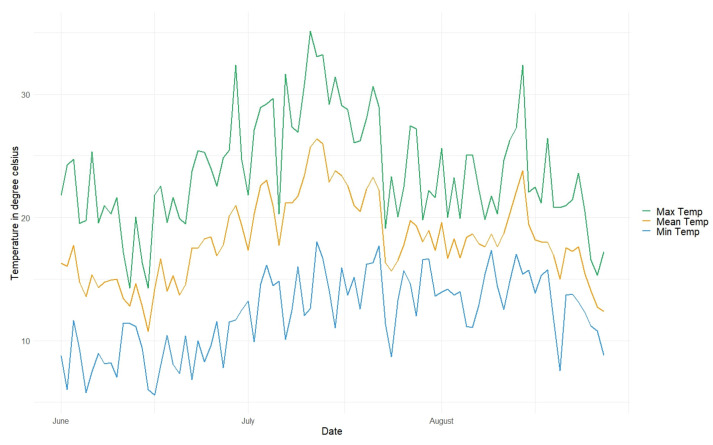
Daily min, mean, and max temperatures recorded throughout the day for the whole experimental period.

**Figure 2 animals-13-03823-f002:**
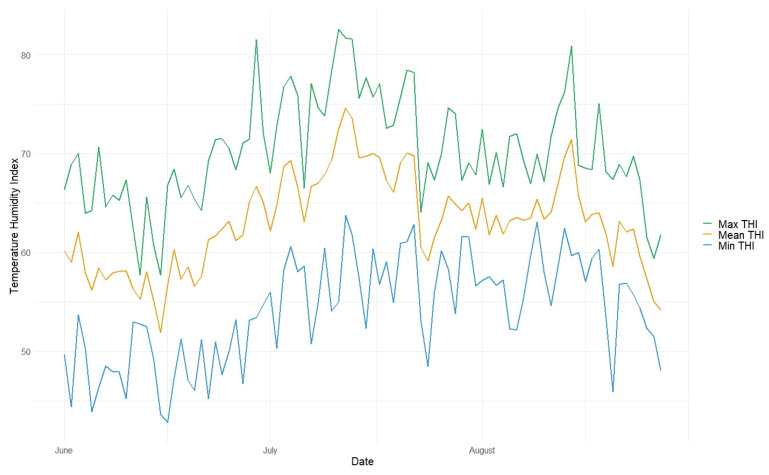
Daily min, mean, and max THI recorded throughout the day for the whole experimental period.

**Table 1 animals-13-03823-t001:** Number of milk yield observations for different lactation stages in treatment groups.

	S Group	NS Group
Stage 1 (DIM ≤ 49)	28	21
Stage 2 (DIM > 49)	837	847

**Table 2 animals-13-03823-t002:** Mean 24 h and daytime (08:00–20:00) temperatures and the number of days with a mean and maximum daytime temperature > 25 °C for the months of June, July, and August.

Month	Mean 24 h Temp. (°C)	Mean Day Temp. (°C)	No. of Days with Mean Temp > 25 °C	No. of Days with Max Temp. > 25 °C
June	15.7 ± 2.4	18.9 ± 3.4	1	4
July	20.9 ± 2.9	24.0 ± 4.0	14	22
August	17.7 ± 2.5	19.9 ± 3.2	2	8

**Table 3 animals-13-03823-t003:** Mean 24 h and daytime (08:00–20:00) THI with standard deviation and the number of days with a mean and maximum daytime THI > 72 for the months of June, July, and August.

Month	Mean 24 h THI	Mean Day THI	No. of Days with Mean Day THI > 72	No. of Days with Max THI > 72
June	59.2 ± 3.3	63.7 ± 4.4	1	2
July	66.9 ± 3.8	70.7 ± 4.8	15	22
August	62.9 ± 3.7	66.1 ± 4.7	2	5

**Table 4 animals-13-03823-t004:** Milk yield data in kg with standard deviation for cows in the different treatment groups in different lactation stages for the months of June, July, and August. Stage 1 = DIM ≤ 49 and stage 2 = DIM > 49.

Month	Shade Group		Non-Shade Group	
	Stage 1	Stage 2	Stage 1	Stage 2
June	37.3 ± 7.7	31.9 ± 8.8	35.8 ± 7.3	32.2 ± 9.4
July	-	32.2 ± 8.0	-	33.1 ± 9.3
August	-	31.1 ± 8.0	-	31.7 ± 8.6

**Table 5 animals-13-03823-t005:** Estimated effects with standard deviation of an increase of 1 °C in the environmental condition on milk yield (kg) and the T- and *p*-values of the three-way interactions between lactation stage, treatment group, and the environmental conditions including mean shade temperature two days before, maximum shade temperature, mean THI two days before, and maximum THI.

Environmental Condition	Treatment Group	Lactation Stage	Estimate	T-Value	*p*-Value
The mean temperature 2 days before	NS	1	−0.36 ± 0.09	t_1669_ = −4.13	<0.001
	S	1	−0.15 ± 0.08	t_1669_ = −2.00	<0.05
	NS	2	−0.06 ± 0.04	t_1669_ =−1.67	ns
	S	2	−0.07 ± 0.04	t_1669_ = −1.60	ns
Maximum temperature	NS	1	−0.22 ± 0.05	t_1707_ = −4.04	<0.001
	S	1	−0.06 ± 0.05	t_1707_ = −1.11	ns
	NS	2	−0.05 ± 0.02	t_1707_ = −2.02	<0.05
	S	2	−0.02 ± 0.02	t_1707_ = −1.05	ns
Mean THI 2 days before	NS	1	−0.14 ± 0.04	t_1669_ = −3.78	<0.001
	S	1	−0.08 ± 0.04	t_1669_ = −2.37	< 0.05
	NS	2	−0.06 ± 0.03	t_1669_ = −2.08	< 0.05
	S	2	−0.06 ± 0.03	t_1669_ = −1.97	< 0.05
Maximum THI	NS	1	−0.10 ± 0.03	t_1707_ = −3.98	<0.001
	S	1	−0.05 ± 0.03	t_1707_ = −1.80	*p* = 0.07
	NS	2	−0.04 ± 0.02	t_1707_ = −2.11	<0.05
	S	2	−0.03 ± 0.02	t_1707_ = −1.45	ns

**Table 6 animals-13-03823-t006:** Effects of environmental conditions on milk production in different lactation stages. Presented are the estimated effects with standard deviation and the T- and *p*-values of the two-way interactions between lactation stage and increases in mean temperature on the day itself, maximum shade temperature two days before, mean THI on the day itself, and maximum THI two days before with milk production for cows in different lactation stages.

	Stage 1			Stage 2		
	Estimate	T-Value	*p*-Value	Estimate	T-Value	*p*-Value
Mean temperature	−0.22 ± 0.06	t_1707_ = −3.87	<0.001	−0.06 ± 0.03	t_1707_ = −2.24	<0.05
Max temperature 2 *	−0.17 ± 0.04	t_1669_ = −4.29	<0.001	−0.03 ± 0.02	t_1669_ = −1.91	*p* = 0.06
Mean THI	−0.10 ± 0.02	t_1707_ = −4.27	<0.001	−0.06 ± 0.02	t_1707_ = −2.88	<0.01
Max THI 2 *	−0.08 ± 0.02	t_1669_ = −4.21	<0.001	−0.03 ± 0.01	t_1669_ = −2.37	<0.05

* 2 = two days before.

## Data Availability

The datasets used for analysing and presenting results for this paper are available upon request from the corresponding author Per Peetz Nielsen (per.peetz.nielsen@ri.se).

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
