# Peer review of "How Does the Provision of Shade during Grazing Affect Heat Stress Experienced by Dairy Cows in Sweden?"

_animals, 2023, doi:10.3390/ani13243823_

Round 1

Reviewer 1 Report

Comments and Suggestions for Authors

Dear authors,

I read your manuscript, the issue is interesting and innovative. 

Notwithstanding this, your experimental scheme shows some oddities.

Introduction

Line 77. Between 12.00 and 14.00 cows with access to shade show decreased grazing behaviour compared to cows without access to shade (19 vs. 44%), resulting from a peak in shade use at that time.

Ok I understand, however I suggest to indicate bibliography.

Lines 82-85. In bibliography [19] you cite the Swedish law, can you add an annex with the main guidelines? Thanks.

Materials and methods

Line 99. Poa pratensis and Festuca pratensis (italic).

Lines 100-102. Can you provide a better description (with a figure or with a scheme if you want) about this stanchion barn (size and materials for example)? Moreover, what do you mean with the sentence: “all other times of the day and night they were outdoors on pasture”? The animals have no shelter?

Line 106. There are some animals completely without shade? It is correct?

Line 122. The temperature was only measured in shade and not under solar radiation. If, as I understand it, (perhaps I’m wrong) a group of animals has no shade, this measurement only affects an experimental group. Thus, the non shade (NS) group cannot be compared with the shade (S) group. If so, I do not understand how the shadow effect can be assessed, being the main concern of the manuscript: “How does the provision of shade during grazing affect heat stress”. Your study is innovative and interesting, however I would need to better understand this aspect of experimental design. Thanks.

Discussion

Line 229. Access to shade seems to decrease the negative effect on milk production. This assumption can be suggested by the common sense, however, as previously indicated, you recorded the temperatures in shade only. Perhaps the study requires a investigation about (in addition to the temperature) the solar irradiance

Lines 239-242. These sentences, these concerns are conclusions, I think.

Conclusions

Your conclusions are negatively affected by the inadequate experimental design, it could not be otherwise. Please can you provide a better description (as still suggested) of your methods? Thanks.

Yours sincerely

Comments on the Quality of English Language

Good quality of English language

Thanks

Author Response

Thank you for reviewing our paper. Your suggestions have improved the paper substantially and we have submitted our answers to your concerns in Italic after your suggestions. 

Introduction

Line 77. Between 12.00 and 14.00 cows with access to shade show decreased grazing behaviour compared to cows without access to shade (19 vs. 44%), resulting from a peak in shade use at that time. Ok I understand, however I suggest to indicate bibliography. Done

Lines 82-85. In bibliography [19] you cite the Swedish law, can you add an annex with the main guidelines? Thanks. I have added the URL for the animal welfare act to the reference. Unfortunately this is in Swedish

Materials and methods

Line 99. Poa pratensis and Festuca pratensis (italic).Done

Lines 100-102. Can you provide a better description (with a figure or with a scheme if you want) about this stanchion barn (size and materials for example)?

Moreover, what do you mean with the sentence: “all other times of the day and night they were outdoors on pasture”? The animals have no shelter? Have changed the text a bit, but no, the cows were out on pasture 24/7 as they would normally be in Sweden

Line 106. There are some animals completely without shade? It is correct? Tried to make it more clear in the text, but yes one group had no shade at all

Line 122. The temperature was only measured in shade and not under solar radiation. If, as I understand it, (perhaps I’m wrong) a group of animals has no shade, this measurement only affects an experimental group. Thus, the non shade (NS) group cannot be compared with the shade (S) group. If so, I do not understand how the shadow effect can be assessed, being the main concern of the manuscript: “How does the provision of shade during grazing affect heat stress”. Your study is innovative and interesting, however I would need to better understand this aspect of experimental design. Thanks. THI is a widely used measure for when cows experience heat stress, and is calculated as described in the text. I deleted the sentence ‘The temperature was only measured in shade and not under solar radiation.’ Since it is not relevant for the study and for calculating when THI reach a lew for heat stress

Discussion

Line 229. Access to shade seems to decrease the negative effect on milk production. This assumption can be suggested by the common sense, however, as previously indicated, you recorded the temperatures in shade only. Perhaps the study requires a investigation about (in addition to the temperature) the solar irradiance. Not relevant when THI is used. Rewrote the sentence to put more focus on Swedish conditions

Lines 239-242. These sentences, these concerns are conclusions, I think. This paragraph is more of a reflection on the limitations that this study have and what should be included in future studies. So I do not consider this part as conclusion.

Conclusions

Your conclusions are negatively affected by the inadequate experimental design, it could not be otherwise. Please can you provide a better description (as still suggested) of your methods? Thanks. We do not agree with this. We are aware of the limitations in the paper, but it still provides a valuable information for future research on heat stress in Scandinavia as the first experiment dealing with this issue in the northern part of Europe.

Reviewer 2 Report

Comments and Suggestions for Authors

Subject: animals-2699706

How does the provision of shade during grazing affect heat stress experienced by dairy cows in Sweden?

General comments:

This study dealing with a serious problem affecting dairy cows around the world – effect of heat stress conditions on dairy cows. This is an interesting issue that a lot of studies deal with. The current study tests the hypothesis that shade can prevent cows for suffering from heat stress in temperate climate like in Sweeden where there is not strong heat stress.

The authors didn’t test rectal temperature so how they can say that the cows were in heat stress. Mean temperature of 20 degree is not consider heat stress.   

The study used very low number of cows (20) and doesn’t look on milk production over time. I think you need to give figure with milk per cow according to time and not just the average. There is big difference in DIM between the groups what can affect milk yield. The statistical model doesn’t take into consideration days in milk during the study. The authors show the negative effect of temp or THI elevation 2 days before measurement on milk yield, then why not to show the effect for longer period? I mean for all summer. The effect of heat stress that we usually see are long and continues effects. This is also the main problem that will create the economical lost.

Specific comments:

Line 23: you didn’t test body temperature or respiration rates so you can not talk about it in the abstract. It belongs to the introduction.

Line 104: using of just 20 cows seems not appropriate for testing milk amount. I think you need more cows to find any effect. Especially when you look on table 1 there is almost no observation during the first period. I think you must add more cows to the study.

Line 107: this is a big difference in the average DIM that can affect milk yield. The author needs to address this point especially in the statistical model.

Lines 138- 146: in the statistical model you wrote that you use the lactation number as main effect and didn’t use the lactation stage as a main effect. You use lactation stage just in interactions with other parameters, while you ignore lactation number in all your analysis and result and there is no data on the effect of lactation number on milk yield, how many cows from each lactation where in the different treatment groups.

Table 2: you have here temperature above the thermos-natural -zone (25Degree) just on July and also not all the month. I think you need to present your result in a figure that will show everyday temperature and just the table with average. The same for THI.

Table 4: if you have cows in stage 1 only in June where there is no heat stress according to your data, how can yo say that there is effect of the shade?

Table 5: you compare between treatments in the same lactation stage so please change the table according: different groups in the same lactation stage and not same group in different stage like it is presented.

Line 222: there is very low number of cows to say this conclusion. Furthermore, in early lactation (June) you still didn’t have heat stress, so I am not sure that the difference are because the weather, maybe they are because the difference in DIM.

Line 230: Again, you repeat on the same thing like in Line 222.

Line 239-245: the author mentions here about the problem of limited number of cows in early lactation and also that there were no cows in early lactation during time with heat stress (July). I think this something they will need to test and to add to the study.     

Author Response

Thank you for reviewing our paper. Your suggestions have improved the paper substantially and we have submitted our answers to your concerns in Italic after your suggestions. 

General comments:

This study dealing with a serious problem affecting dairy cows around the world – effect of heat stress conditions on dairy cows. This is an interesting issue that a lot of studies deal with. The current study tests the hypothesis that shade can prevent cows for suffering from heat stress in temperate climate like in Sweeden where there is not strong heat stress.

The authors didn’t test rectal temperature so how they can say that the cows were in heat stress. Mean temperature of 20 degree is not consider heat stress.   We do not use mean temperature as an sole indication of heat stress, but the well validated THI. We are aware of that temperature in Sweden do not reach Saudi Arabia temperature, but still find it extremely interesting to test if even Swedish cows are affected by an increase in THI.

The study used very low number of cows (20) and doesn’t look on milk production over time. I think you need to give figure with milk per cow according to time and not just the average. There is big difference in DIM between the groups what can affect milk yield. The statistical model doesn’t take into consideration days in milk during the study. The authors show the negative effect of temp or THI elevation 2 days before measurement on milk yield, then why not to show the effect for longer period? I mean for all summer. The effect of heat stress that we usually see are long and continues effects. This is also the main problem that will create the economical lost. We do not agree that 20 cows in each group is a low number, especially when shade needs to be provided with a structure as often used in scientific experiments. We show a more acute effect of a high THI since this is Sweden and having a month with 30+ degrees is not as common as in a much warmer climate. Therefore, it is not so interesting to look at long-term effects.

Specific comments:

Line 23: you didn’t test body temperature or respiration rates so you can not talk about it in the abstract. It belongs to the introduction. It is the introduction in the abstract, building up a case for why heat stress is of interest.

Line 104: using of just 20 cows seems not appropriate for testing milk amount. I think you need more cows to find any effect. Especially when you look on table 1 there is almost no observation during the first period. I think you must add more cows to the study. We agree that we have a small number of cows in early lactation, and as we state in the end of the discussion, more studies are needed but this study is the first studi under Nordic conditions and are of interest to the scientific community.

Line 107: this is a big difference in the average DIM that can affect milk yield. The author needs to address this point especially in the statistical model. We are not looking at milk production between cows or changes in milk production over a long period, therefore, the cows act as its own control and DIM is not included in the statistical analysis.

Lines 138- 146: in the statistical model you wrote that you use the lactation number as main effect and didn’t use the lactation stage as a main effect. You use lactation stage just in interactions with other parameters, while you ignore lactation number in all your analysis and result and there is no data on the effect of lactation number on milk yield, how many cows from each lactation where in the different treatment groups. Thanks for spotting this, our mistake. Have now been added in the text.

Table 2: you have here temperature above the thermos-natural -zone (25Degree) just on July and also not all the month. I think you need to present your result in a figure that will show everyday temperature and just the table with average. The same for THI. Since this experiment focuses on a more acute effect of a high THI and not a high THI over time, this information will not add any valuable information to this paper.

Table 4: if you have cows in stage 1 only in June where there is no heat stress according to your data, how can yo say that there is effect of the shade? A THI level of 72 as a cut of point is up for debate, and this experiment show that even when there really is not any days with THI above 72, cows early in lactation can be affected. This is why we discuss this effect even with the low number of cows. Furthermore as described in the paper, we did not want to change cows during the experiment, with the effect that early stage could more or less only occur in June.

Table 5: you compare between treatments in the same lactation stage so please change the table according: different groups in the same lactation stage and not same group in different stage like it is presented. Thanks for pointing this out. Have changed the text according to your suggestion.

Line 222: there is very low number of cows to say this conclusion. Furthermore, in early lactation (June) you still didn’t have heat stress, so I am not sure that the difference are because the weather, maybe they are because the difference in DIM. We do not agree with this comment. The results show that even with a low number of days with a THI above 72, milk production is affected. We do  not conclude that cows in early lactation benefit from have access to shade in our study, and we even discuss in the discussion that the low number of cows in early lactation is a limit of this exoperiment.

Line 230: Again, you repeat on the same thing like in Line 222. This effect we see might be due to the fact that the cut of point of THI=72 might be too high for high producing cows. Our results show that providing shade affect milk production during the summer.

Line 239-245: the author mentions here about the problem of limited number of cows in early lactation and also that there were no cows in early lactation during time with heat stress (July). I think this something they will need to test and to add to the study. We agree and will seek funding for repeating this experiment in the future, especially with the last years increase in thye global temperature, heat stress have be brought to the agenda even in Sweden.

Reviewer 3 Report

Comments and Suggestions for Authors

Dear authors,

After carefully reading your interesting manuscript on the effects of heat stress on dairy cattle milk production, I have a set of suggestions for improving the quality of your article:

- Lines 18-22: Use the past tense when presenting own results/findings;

- L 26: Please replace 'high lactating' with 'high producing or yielding cows', and throughout the text;

- L 30-31: Please correct the statement 'two days ago' and throughout the manuscript, I am not sure what the authors meant by this;

- L 30-32: the high THI has affected the THI? Please correct this statement or delete it;

- L 76: please include 'grazing behavioural pattern' or 'grazing circadian rhythm';

- Please explain why the study was conduced in 2010, and you are trying to have it published only 13 years later. Surly this simple data to evaluate did not involved any complicated formal analysis and such;

- Line 252: replace 'help save milk' with 'improve milk production/milk yields';

- Given the very short manuscript, the experimental design that is not extremely complex and also the limited parameters (basically temperature and milk yield) studied, I recommend the publication of the manuscript as a 'Short Communication', rather than a 'Regular Article'.

Author Response

Thank you for reviewing our paper. Your suggestions have improved the paper substantially and we have submitted our answers to your concerns in Italic after your suggestions. 

After carefully reading your interesting manuscript on the effects of heat stress on dairy cattle milk production, I have a set of suggestions for improving the quality of your article:

- Lines 18-22: Use the past tense when presenting own results/findings; Thanks for pointing this out, the text have now been changed to past tense

- L 26: Please replace 'high lactating' with 'high producing or yielding cows', and throughout the text; Done

- L 30-31: Please correct the statement 'two days ago' and throughout the manuscript, I am not sure what the authors meant by this; ‘Two days ago’ is used to describe the two day lac in effect of heat stress on milk production and can be changed with ‘two days previous’ but the later can sound a bit to formal and is not commonly used.

- L 30-32: the high THI has affected the THI? Please correct this statement or delete it; We cannot find this in the text so cannot correct it.

- L 76: please include 'grazing behavioural pattern' or 'grazing circadian rhythm'; Added

- Please explain why the study was conduced in 2010, and you are trying to have it published only 13 years later. Surly this simple data to evaluate did not involved any complicated formal analysis and such; Sometimes life happens and other stuff come in the way. This paper have unfortunately been lying on our desk for far to long. Bet better late than never!

- Line 252: replace 'help save milk' with 'improve milk production/milk yields'; Done

- Given the very short manuscript, the experimental design that is not extremely complex and also the limited parameters (basically temperature and milk yield) studied, I recommend the publication of the manuscript as a 'Short Communication', rather than a 'Regular Article'. This is the first paper on heat stress under Scandinavian conditions and provides value information to future research on heat stress in the region. We are aware of the limitations presented in this paper, but still think it has value as a full paper.

Round 2

Reviewer 1 Report

Comments and Suggestions for Authors

Line 78. I think that ambient temperature should be changed with environmental temperature. 

Line 118. Please, you should change the cows was fletched " with "the cows were fletched".

Above, some examples of sentences that should be improved. Please check the manuscript (english grammar), afterwards the manuscript could be accepted

In the discussion. I suggested to consider some sentences as conclusions, you disagree. Notwithstanding this, as peer reviewer I must respect your approach.

Sincerely

Comments on the Quality of English Language

English is overall good

Author Response

Line 78. I think that ambient temperature should be changed with environmental temperature. Done

Line 118. Please, you should change the cows was fletched " with "the cows were fletched". Done

Above, some examples of sentences that should be improved. Please check the manuscript (english grammar), afterwards the manuscript could be accepted We have used Grammarly throughout the writing process to check and improve the language and grammar. Have been through the manuscript once again and found some more errors that have now been corrected. Thanks for highlighting that even AI is not perfect ?

In the discussion. I suggested to consider some sentences as conclusions, you disagree. Notwithstanding this, as peer reviewer I must respect your approach.

Reviewer 2 Report

Comments and Suggestions for Authors

dear authhor's, 

in green you will see my additional comments.

author's Notes

Thank you for reviewing our paper. Your suggestions have improved the paper substantially and we have submitted our answers to your concerns in Italic after your suggestions. 

General comments:

This study dealing with a serious problem affecting dairy cows around the world – effect of heat stress conditions on dairy cows. This is an interesting issue that a lot of studies deal with. The current study tests the hypothesis that shade can prevent cows for suffering from heat stress in temperate climate like in Sweeden where there is not strong heat stress.

The authors didn’t test rectal temperature so how they can say that the cows were in heat stress. Mean temperature of 20 degree is not consider heat stress.

 We do not use mean temperature as an sole indication of heat stress, but the well validated THI. We are aware of that temperature in Sweden do not reach Saudi Arabia temperature, but still find it extremely interesting to test if even Swedish cows are affected by an increase in THI.

R: this is interesting to look on the effect even in Sweden where there is not strong heat stress but I still think that looking on rectal temperature during the all day will add more reliable data

The study used very low number of cows (20) and doesn’t look on milk production over time. I think you need to give figure with milk per cow according to time and not just the average. There is big difference in DIM between the groups what can affect milk yield. The statistical model doesn’t take into consideration days in milk during the study. The authors show the negative effect of temp or THI elevation 2 days before measurement on milk yield, then why not to show the effect for longer period? I mean for all summer. The effect of heat stress that we usually see are long and continues effects. This is also the main problem that will create the economical lost.

We do not agree that 20 cows in each group is a low number, especially when shade needs to be provided with a structure as often used in scientific experiments. We show a more acute effect of a high THI since this is Sweden and having a month with 30+ degrees is not as common as in a much warmer climate. Therefore, it is not so interesting to look at long-term effects.

R: I still not agree with your point, today with the increase in global temperature it will be interesting to test long term effects that are more important from short term effect. At least discus it in your MS. 

Specific comments:

Line 23: you didn’t test body temperature or respiration rates so you can not talk about it in the abstract. It belongs to the introduction

It is the introduction in the abstract, building up a case for why heat stress is of interest.

R: ok.

Line 104: using of just 20 cows seems not appropriate for testing milk amount. I think you need more cows to find any effect. Especially when you look on table 1 there is almost no observation during the first period. I think you must add more cows to the study.

We agree that we have a small number of cows in early lactation, and as we state in the end of the discussion, more studies are needed but this study is the first study under Nordic conditions and are of interest to the scientific community.

R: yes, but still to have strong result you need more cows in each group. This is to low number to talk about so strong effect. You need to give more attention to this point.

Line 107: this is a big difference in the average DIM that can affect milk yield. The author needs to address this point especially in the statistical model.

We are not looking at milk production between cows or changes in milk production over a long period, therefore, the cows act as its own control and DIM is not included in the statistical analysis.

R: cows can't serve as there own control because there is DIM effect on milk yield. We know that there is reduction after peak yield every week so this is very strong point that can affect your analysis.

Lines 138- 146: in the statistical model you wrote that you use the lactation number as main effect and didn’t use the lactation stage as a main effect. You use lactation stage just in interactions with other parameters, while you ignore lactation number in all your analysis and result and there is no data on the effect of lactation number on milk yield, how many cows from each lactation where in the different treatment groups.

Thanks for spotting this, our mistake. Have now been added in the text.

R: thank you. But now you see the low number of cows in each lactation. You can't talk about primiparous cows and multiparous cows as the same cows. There is big different in milk production between lactation number. So this is a big problem you didn’t address in your study.

Table 2: you have here temperature above the thermos-natural -zone (25Degree) just on July and also not all the month. I think you need to present your result in a figure that will show everyday temperature and just the table with average. The same for THI.

Since this experiment focuses on a more acute effect of a high THI and not a high THI over time, this information will not add any valuable information to this paper.

R: I don't agree. I think this is important like I think it is important to show the rectal temperature during 24h.

Table 4: if you have cows in stage 1 only in June where there is no heat stress according to your data, how can you say that there is effect of the shade? 

A THI level of 72 as a cut of point is up for debate, and this experiment show that even when there really is not any days with THI above 72, cows early in lactation can be affected. This is why we discuss this effect even with the low number of cows. Furthermore as described in the paper, we did not want to change cows during the experiment, with the effect that early stage could more or less only occur in June.

R: for my opinion those cows in June didn’t suffer from heat stress according to your data. that why it is important to present the rectal temperature data. this is very important issue that create problem during the all study.

Table 5: you compare between treatments in the same lactation stage so please change the table according: different groups in the same lactation stage and not same group in different stage like it is presented.

Thanks for pointing this out. Have changed the text according to your suggestion.

R: you didn’t change the table.

This is what I mean: you compare in the same stage between groups so this is the wright way to create the table it:

Environmental condition

Treatment group

Lactation stage

Estimate

T-value

P-value

Mean temperature 2 days ago

NS

1

S

1

NS

2

S

2

Line 222: there is very low number of cows to say this conclusion. Furthermore, in early lactation (June) you still didn’t have heat stress, so I am not sure that the difference are because the weather, maybe they are because the difference in DIM. 

We do not agree with this comment. The results show that even with a low number of days with a THI above 72, milk production is affected. We do  not conclude that cows in early lactation benefit from have access to shade in our study, and we even discuss in the discussion that the low number of cows in early lactation is a limit of this exoperiment.

R: see my previous comment on this issue.

Line 230: Again, you repeat on the same thing like in Line 222. 

This effect we see might be due to the fact that the cut of point of THI=72 might be too high for high producing cows. Our results show that providing shade affect milk production during the summer.

Line 239-245: the author mentions here about the problem of limited number of cows in early lactation and also that there were no cows in early lactation during time with heat stress (July). I think this something they will need to test and to add to the study. 

We agree and will seek funding for repeating this experiment in the future, especially with the last years increase in thye global temperature, heat stress have be brought to the agenda even in Sweden.

R: I think you need to add this to current MS.  

Author Response

Thank you for reviewing our paper. Your suggestions have improved the paper substantially and we have submitted our answers to your concerns in Italic after your suggestions. 

General comments:

This study dealing with a serious problem affecting dairy cows around the world – effect of heat stress conditions on dairy cows. This is an interesting issue that a lot of studies deal with. The current study tests the hypothesis that shade can prevent cows for suffering from heat stress in temperate climate like in Sweeden where there is not strong heat stress.

The authors didn’t test rectal temperature so how they can say that the cows were in heat stress. Mean temperature of 20 degree is not consider heat stress.

 We do not use mean temperature as an sole indication of heat stress, but the well validated THI. We are aware of that temperature in Sweden do not reach Saudi Arabia temperature, but still find it extremely interesting to test if even Swedish cows are affected by an increase in THI.

R: this is interesting to look on the effect even in Sweden where there is not strong heat stress but I still think that looking on rectal temperature during the all day will add more reliable data

We can only agree that adding body temperature would have been interesting for this study. The first author used temperature loggers for doing this when working with heat stress at AgReseacrh in New Zealand, but we also need to acknowledge that the nights in Sweden so far are not as warm as in e.g New Zealand, and therefore the cows have a greater possibility to have a decrease in body temperature due to the cooling effect of low temperature during the night. However, if we get further funding for examining heat stress in dairy cattle in Sweden we will do our best to include body temperature as well.

The study used a very low number of cows (20) and doesn’t look on milk production over time. I think you need to give figure with milk per cow according to time and not just the average. There is big difference in DIM between the groups what can affect milk yield. The statistical model doesn’t take into consideration days in milk during the study. The authors show the negative effect of temp or THI elevation 2 days before measurement on milk yield, then why not to show the effect for longer period? I mean for all summer. The effect of heat stress that we usually see are long and continues effects. This is also the main problem that will create the economical lost.

We do not agree that 20 cows in each group is a low number, especially when shade needs to be provided with a structure as often used in scientific experiments. We show a more acute effect of a high THI since this is Sweden and having a month with 30+ degrees is not as common as in a much warmer climate. Therefore, it is not so interesting to look at long-term effects.

R: I still not agree with your point, today with the increase in global temperature it will be interesting to test long term effects that are more important from short term effect. At least discus it in your MS. 

We agree that it would be interesting to look at longer term effects of heat stress. But unfortunately, we do not yet have 3-6 months of THI above 72 or even a month in a row, so it is not yet possible to do this. Therefore, we thought that it would be interesting to examine the acute effect of a short period of warm weather just to examine if heat stress can be seen as a problem in Sweden. Now we know that cows here can have problems with heat stress and we just need to get more funding and ‘hope’ for increased climate changes to be able to repeat the study and test for long term effect. We do not see that adding a discussion about acute or long term effect of heat stress will ad value to this paper since we do only look at the acute effect of warm weather.

Specific comments:

Line 23: you didn’t test body temperature or respiration rates so you can not talk about it in the abstract. It belongs to the introduction

It is the introduction in the abstract, building up a case for why heat stress is of interest.

R: ok.

Line 104: using of just 20 cows seems not appropriate for testing milk amount. I think you need more cows to find any effect. Especially when you look on table 1 there is almost no observation during the first period. I think you must add more cows to the study.

We agree that we have a small number of cows in early lactation, and as we state in the end of the discussion, more studies are needed but this study is the first study under Nordic conditions and are of interest to the scientific community.

R: yes, but still to have strong result you need more cows in each group. This is to low number to talk about so strong effect. You need to give more attention to this point.

We cannot see how we can give more attention to this point. At the end of the discussion, we state that more studies are needed with more cows since we have a limited amount of data. Unfortunately, we cannot add more data/cows, but with this publication, we might be able to show the funding agencies in Sweden that this indeed is an issue here but more data is needed to draw stronger conclusions and to give recommendations for dairy farmers.

Line 107: this is a big difference in the average DIM that can affect milk yield. The author needs to address this point especially in the statistical model.

We are not looking at milk production between cows or changes in milk production over a long period, therefore, the cows act as its own control and DIM is not included in the statistical analysis.

R: cows can't serve as there own control because there is DIM effect on milk yield. We know that there is reduction after peak yield every week so this is very strong point that can affect your analysis.

Sorry, our mistake. Cow is indeed not used as its own control but in the statistical analysis, she is added as a repeated factor with and AR1 structure. Added this information in the text L171-172

Lines 138- 146: in the statistical model you wrote that you use the lactation number as main effect and didn’t use the lactation stage as a main effect. You use lactation stage just in interactions with other parameters, while you ignore lactation number in all your analysis and result and there is no data on the effect of lactation number on milk yield, how many cows from each lactation where in the different treatment groups.

Thanks for spotting this, our mistake. Have now been added in the text.

R: thank you. But now you see the low number of cows in each lactation. You can't talk about primiparous cows and multiparous cows as the same cows. There is big different in milk production between lactation number. So this is a big problem you didn’t address in your study.

We agree that there is a big difference in milk production between cows and lactation numbers and this is also why cow is used as a repeated measure in the statistical test. We are not interested to look at differences or effect of heat stress between cows, but only within cow. It would have been nice to have a higher number of cows in the study, but since this study was performed at a research farm, with the higher quality of data, the budget in the project could not support more animals. We address the low number of cows in the discussion.

Table 2: you have here temperature above the thermos-natural -zone (25Degree) just on July and also not all the month. I think you need to present your result in a figure that will show everyday temperature and just the table with average. The same for THI.

Since this experiment focuses on a more acute effect of a high THI and not a high THI over time, this information will not add any valuable information to this paper.

R: I don't agree. I think this is important like I think it is important to show the rectal temperature during 24h.

We have added the information about temperature and THI for the whole period in figure 1 and 2. We agree that rectal temperature would have been a nice to have and will consider adding this to our next experiment if funding is available since it will give valuable information about the animals' ability to cope with the heat.

Table 4: if you have cows in stage 1 only in June where there is no heat stress according to your data, how can you say that there is effect of the shade? 

A THI level of 72 as a cut of point is up for debate, and this experiment show that even when there really is not any days with THI above 72, cows early in lactation can be affected. This is why we discuss this effect even with the low number of cows. Furthermore as described in the paper, we did not want to change cows during the experiment, with the effect that early stage could more or less only occur in June.

R: for my opinion those cows in June didn’t suffer from heat stress according to your data. that why it is important to present the rectal temperature data. this is very important issue that create problem during the all study.

We agree that rectal temperature would have been nice to have. But it is not possible for us to add this now since we do not have it. Furthermore, the first author has experience with measuring body temperature inter vaginally and these loggers cannot stay for more than 7-10 days before the cow needs to have at least a week's rest from the logger. So might not have added anything to this study since we do not have a long period of really warm weather.

Table 5: you compare between treatments in the same lactation stage so please change the table according: different groups in the same lactation stage and not same group in different stage like it is presented.

Thanks for pointing this out. Have changed the text according to your suggestion.

R: you didn’t change the table.

This is what I mean: you compare in the same stage between groups so this is the wright way to create the table it:

Environmental condition

Treatment group

Lactation stage

Estimate

T-value

P-value

Mean temperature 2 days ago

NS

1

S

1

NS

2

S

2

 Sorry, misunderstood your point, the table have now been changed as per your suggestion

Line 222: there is very low number of cows to say this conclusion. Furthermore, in early lactation (June) you still didn’t have heat stress, so I am not sure that the difference are because the weather, maybe they are because the difference in DIM. 

We do not agree with this comment. The results show that even with a low number of days with a THI above 72, milk production is affected. We do  not conclude that cows in early lactation benefit from have access to shade in our study, and we even discuss in the discussion that the low number of cows in early lactation is a limit of this exoperiment.

R: see my previous comment on this issue.

We used the cow as a repeated statement in the statistical analysis with an autoregressive correlation structure so the model takes into account that what is interesting to test is milk production from day to day for each cow, not comparing between lactation numbers or DIM.

Line 230: Again, you repeat on the same thing like in Line 222. 

This effect we see might be due to the fact that the cut of point of THI=72 might be too high for high producing cows. Our results show that providing shade affect milk production during the summer.

Line 239-245: the author mentions here about the problem of limited number of cows in early lactation and also that there were no cows in early lactation during time with heat stress (July). I think this something they will need to test and to add to the study. 

We agree and will seek funding for repeating this experiment in the future, especially with the last years increase in thye global temperature, heat stress have be brought to the agenda even in Sweden.

R: I think you need to add this to current MS.  

This is being discussed in the discussion we do not agree that it should be in the method section.

Reviewer 3 Report

Comments and Suggestions for Authors

Thank you for making the effort to revise your manuscript, while taking into account some of my recommendations/comments. 

Author Response

Thank you for your valuable comments

Round 3

Reviewer 2 Report

Comments and Suggestions for Authors

thank you for the answers and adding the data and corrected statistical analasis and table. 

one thing: in figure 1 you need to correct the heading to Y axis that it will be "temprature in celsiuis degree and like it iw written now.

 I dont have any additional commenets,